# Influence of the Infill Orientation on the Properties of Zirconia Parts Produced by Fused Filament Fabrication

**DOI:** 10.3390/ma13143158

**Published:** 2020-07-15

**Authors:** Santiago Cano, Tanja Lube, Philipp Huber, Alberto Gallego, Juan Alfonso Naranjo, Cristina Berges, Stephan Schuschnigg, Gemma Herranz, Christian Kukla, Clemens Holzer, Joamin Gonzalez-Gutierrez

**Affiliations:** 1Department of Polymer Engineering and Science, Institute of Polymer Processing, Montanuniversitaet Leoben, Otto Gloeckel-Straße 2, 8700 Leoben, Austria; philipp-arno-franz.huber@stud.unileoben.ac.at (P.H.); stephan.schuschnigg@unileoben.ac.at (S.S.); clemens.holzer@unileoben.ac.at (C.H.); joamin.gonzalez-gutierrez@unileoben.ac.at (J.G.-G.); 2Department of Materials Science, Chair of Structural and Functional Ceramics, Montanuniversitaet Leoben, Franz Josef-Straße 18, 8700 Leoben, Austria; 3INEI-ETSII, University of Castilla-La Mancha, Av. Camilo José Cela s/n, 13071 Ciudad Real, Spain; alberto.gallego@alu.uclm.es (A.G.); juanalfonso.naranjo@uclm.es (J.A.N.); cristina.berges@uclm.es (C.B.); gemma.herranz@uclm.es (G.H.); 4Industrial Liaison, Montanuniversitaet Leoben, Franz-Josef-Straße 18, 8700 Leoben, Austria; Christian.kukla@unileoben.ac.at

**Keywords:** fused filament fabrication, zirconia, material extrusion, infill orientation, raster orientation, fill pattern, rheology, bending strength, printing defects

## Abstract

The fused filament fabrication (FFF) of ceramics enables the additive manufacturing of components with complex geometries for many applications like tooling or prototyping. Nevertheless, due to the many factors involved in the process, it is difficult to separate the effect of the different parameters on the final properties of the FFF parts, which hinders the expansion of the technology. In this paper, the effect of the fill pattern used during FFF on the defects and the mechanical properties of zirconia components is evaluated. The zirconia-filled filaments were produced from scratch, characterized by different methods and used in the FFF of bending bars with infill orientations of 0°, ±45° and 90° with respect to the longest dimension of the specimens. Three-point bending tests were conducted on the specimens with the side in contact with the build platform under tensile loads. Next, the defects were identified with cuts in different sections. During the shaping by FFF, pores appeared inside the extruded roads due to binder degradation and or moisture evaporation. The changes in the fill pattern resulted in different types of porosity and defects in the first layer, with the latter leading to earlier fracture of the components. Due to these variations, the specimens with the 0° infill orientation had the lowest porosity and the highest bending strength, followed by the specimens with ±45° infill orientation and finally by those with 90° infill orientation.

## 1. Introduction

Thanks to the considerable efforts and interest of the industry and research institutions, the ceramic additive manufacturing has experienced considerable development and growth that is forecasted to increase in the next years [1]. Among the many technologies that have been developed in the last years [2,3,4,5,6], the four most relevant process categories are [1] vat photopolymerization (VPP), binder jetting (BJT), material jetting (MJT) and material extrusion (MEX). Each additive manufacturing technique has different advantages and disadvantages in terms of speed, dimensional accuracy, properties of the final components and costs [2,3,4,5,6,7]. Therefore, the feasibility of using one technique or another depends on the application.

The fused filament fabrication (FFF) of ceramics, also known as Fused Deposition of Ceramics (FDC) [8,9,10], is a MEX process based on the selective extrusion through a nozzle of feedstock in the shape of filament. The feedstock is a multicomponent polymer system highly filled with powder of the desired ceramic material [9,11,12,13]. Once the so-called green parts are shaped by FFF, the polymer components are removed in the debinding stage by a catalytic reaction [14], dissolution in a solvent [12,15,16] and/or thermal decomposition [9,17,18,19]. Finally, the parts are sintered to obtain nearly dense components. Some advantages of this technology include the simplicity and the low cost of the equipment required to shape the parts [20], the possibility to produce lightweight structures with closed geometry [19,20,21], the possibility to combine this technology with a well-established process like ceramic injection molding [22], and the ability to combine various materials in one part by the use of various nozzles [23,24,25]. Nevertheless, the use of filaments as feed materials requires tight dimensional tolerances of the filaments to ensure proper feeding [8,26,27], which increases its price [28]. Moreover, the roughness in FFF parts and the minimum dimensions achievable are limited by the diameter of the nozzle [29,30].

In FFF and the other MEX techniques [8,14,31,32,33,34,35,36], the generation of the shape by the extrusion of material strands, known as roads, can result in defects in the gaps between the strands or weak areas if the bonding between the strands is not sufficient [8]. In early investigations, different solutions were proposed to reduce these defects [8,10,37]. However, defects persist and are the cause of anisotropic mechanical properties even for nearly dense ceramics produced by other MEX techniques [38].

These typical FFF shaping defects depend on different factors such as the properties of the material, the processing parameters, the orientation of the parts with respect to the build platform and the pattern followed in the deposition of the material in each layer [39]. In most cases, one or more perimeters are included in each layer following the contour of the part to improve the surface quality, and a raster fill pattern is used in the inner part of the specimens, known as the infill. The orientation of the roads in the infill of the parts is known to have a substantial effect on the properties of polymeric components produced by FFF [37,39,40], and the first studies for metallic FFF components showing similar effects are also available [14]. In the case of FFF ceramic components, the infill orientation is known to have a small effect on the dimensional variations during debinding [17]. However, no detailed investigation is available for ceramic parts on the effect of the infill orientation on the appearance of defects during the shaping by FFF and the mechanical properties of the final components.

Hence, the objective of this work was to study the influence of the infill orientation on the properties of zirconia FFF components. The feedstock filament was produced from scratch, characterized by different methods and used in the FFF of specimens with infill orientations of 0°, 90° and ±45° with respect to the longest dimension of the part. The causes of the defects of the final specimens are related to the feedstock properties and the fill pattern used in the FFF with the different infill orientations. Finally, the influence of the defects on the bending properties of the various specimens is discussed.

## 2. Materials and Methods

### 2.1. Materials

The powder used was the tetragonal zirconia (ZrO_2_)—stabilized with 3 mol % yttria (Y_2_O_3_) TZ-3YS-E (Tosoh Europe B.V., Amsterdam, The Netherlands), supplied as spray-dried granules. According to the supplier, the powder has an average particle size of 90 nm and a specific surface area of 7 ± 2 m^2^·g^−1^ [41]. The powder was mixed with the multicomponent binder system developed at the Institute of Polymer Processing of the Montanuniversitaet Leoben. This system is composed of a commercial thermoplastic elastomer compound (TPE, Kraiburg TPE GmbH & Co. KG, Waldkraiburg, Germany) and polyolefin grafted with a polar component to improve the adhesion to the powder (gPO, BYK Chemie GmbH, Wesel, Germany). The specific polymer grades employed and the fraction of each binder component are confidential.

### 2.2. Feedstock and Filament Production

The first step in the production of the ceramic filaments was the compounding of the binder system. Pellets of the two components of the binder system were fed together and mixed in the co-rotating twin-screw compounder ZSK 25 (Werner & Pfleiderer GmbH, Stuttgart, Germany). The rotational speed was 270 rpm, and the compounder had a temperature profile from the feed zone to the nozzle of 120, 185, 190, 195 and 200 °C. The material coming out of the nozzle was cooled down with water at room temperature and pelletized using a water bath + pelletizer (Accrapak Systems Ltd., Burtonwood, United Kingdom). An air blade removed most of the water from the surface of the material entering the pelletizer. In order to remove the remaining water, the binder pellets were dried overnight at 60 °C with a hot, dry air drier (Wittmann Kunststoffgeräte GmbH, Vienna, Austria).

The binder system was compounded with the zirconia powder to obtain a feedstock with 47 vol% of powder, which is equivalent to 85.3 wt%. The feedstock was produced in the co-rotating twin-screw compounder Leistritz ZSE 18 HP-48D (Leistritz Extrusionstechnik GmbH, Nuremberg, Germany), equipped with a high-shear screw configuration to ensure the homogeneous dispersion of the ceramic particles in the polymer. The screw speed was set to 600 rpm, and the total mass output (powder + binder) was set to 6 kg·h^−1^. The binder was introduced in the first zone of the compounder, and the powder was added in the fourth zone using a side feeder consisting of co-rotating screws rotating at 250 rpm. The temperature of the first zone was set to 25 °C and progressively increased from the second zone to the nozzle with the profile: 180, 200, 205, 205, 210, 210, 210, 210, 210, 210, 220 and 220 °C. Once compounded, the molten feedstock coming out of the nozzle was cooled down on an air-cooled metal conveyor belt and pelletized (Reduction Engineering Scheer, Kent, OH, USA).

The feedstock pellets were then used in the production of filaments in the single screw extruder FT-E20T-MP-IS (Dr Collin GmbH, Ebersberg, Germany) equipped with a die of 1.75 mm diameter. The extrusion temperatures from the feed zone to the nozzle were 230, 235, 240 and 245 °C and the screw rotation speed was set to 19 rpm. A polytetrafluoroethylene (PTFE) conveyor belt collected the extruded filament, which cooled down by natural convection. A self-developed haul-off and spooling unit was used to spool the filaments, whose diameter was measured with the laser measuring device, Diagnostic Laser 2000 (SIKORA AG, Bremen, Germany).

### 2.3. Feedstock Evaluation

The morphology of the cryofractured filaments was studied by scanning electron microscopy (SEM, Tescan Vega II, Tescan Brno, s.r.o., Czech Republic). The analyses were performed on gold-sputtered (100 s at 20 mA) specimens at 5 kV using secondary electrons.

Tensile properties of the filaments were measured on 100 mm long straight specimens using the universal testing machine Zwick Z001 (Zwick GmbH & Co.KG, Ulm, Germany) with a 1 kN load cell and pneumatic grips. An initial gauge length of 50 mm was set for all the measurements. The tests, five repetitions each, were performed at standardized conditions (23 °C and 50% relative humidity), at a speed of 10 mm·min^−1^ until rupture.

The rheological behavior of the feedstock was characterized using regranulated filaments. The binder obtained in the first compounding step was also evaluated for comparative purposes. Capillary rheology measurements of both materials were conducted in the high-pressure capillary rheometer Rheograph 2002 (Göttfert Werkstoff-Prüfmaschinen GmbH, Buchen, Germany). The tests were conducted at a temperature of 255 °C and apparent shear rates from 75 to 750 s^−1^. Three round dies with a diameter of 1 mm and lengths of 10, 20 and 30 mm were employed. For the binder system, three tests were carried out with each nozzle, whereas five measurements per die were conducted for the feedstock due to the high variations in the recorded pressure. The true shear rate and viscosity values were calculated with the Bagley [42] and Weisenberg–Rabinowitsch [43,44] corrections, respectively.

Thermogravimetric analysis (TGA) of the binder and the feedstock filament were conducted in the TGA/DSC1 (Mettler-Toledo GmbH, Greifensee, Switzerland) under an oxygen atmosphere. The tests were conducted from 25 °C to 600 °C with a heating rate of 10 K·min^−1^. Alumina crucibles were used, and a total of 5 measurements per material were conducted.

### 2.4. Sample Production

Prismatic specimens with a length of 40.55 mm, a width of 3.93 mm and a thickness of 3.6 mm were produced using FFF. Figure 1 shows the fill pattern in one layer for each of the infill orientations. This fill pattern is the same in all the layers of the specimens with the 0° and 90° infill orientations. In contrast, in the ±45° infill orientation, the roads of the infill are perpendicular to the roads of the previous layer. Specimen dimensions were set considering a homogeneous shrinkage of 20% and adjusting the length and width so that a whole number of roads were extruded without gaps in the 90° and 0° infill orientations, respectively. The software Simplify3D version 4.1.2 (Simplify3D, Blue Ash, OH, USA) was used to slice the parts and generate the G-Code. In Appendix A the G-Codes for the 0°, ±45° and 90° orientations can be found. The parts were produced using the Duplicator i3 v2 (Wanhao, Jinhua, Zhejiang, China) FFF machine. A brass nozzle with TwinClad^®^ XT coating with a diameter of 0.6 mm and a PTFE tube insert was used. The set layer thickness was 0.15 mm for all the layers, and the printing speed was 5 mm·s^−1^ for the first layer and 12.5 mm·s^−1^ for the rest of the layers. Independently of the infill orientation, one perimeter was printed with an infill-perimeter overlap of 50% and an extrusion multiplier of 85%. The build platform was a glass mirror coated with hair spray for better adhesion. The extruder and build platform temperatures were set to 255 °C and 100 °C, respectively. Before starting the specimen production, the build platform and the extruder were preheated for 30 min; the distance of the die to the build platform was calibrated in the printing area. Then, the hair spray coating was applied, and the printing started. The parts were printed in build cycles of 5 specimens per print, with a total of 4 build cycles for the 0° and ±45° infill orientations and 5 build cycles for the 90° infill orientation.

A two-step debinding process was carried out. First, most of the TPE was dissolved in cyclohexane. Solvent debinding in cyclohexane was performed in the digital thermostatic bath (Ovan B105-DE, Barcelona, Spain). The parts were immersed in cyclohexane at 60 °C for 24 h and 12 mL of fresh solvent per gram of feedstock were used. After 24 h in cyclohexane, the specimens were left to dry in the LAN Technics stove (Labolan, Esparza, Spain) at 50 °C for 24 h. Subsequently, the thermal debinding was conducted in the Hobersal furnace 12PR450/SCH PAD P (Hobersal, Barcelona, Spain) in an air atmosphere. The thermal debinding cycle employed was: heating from room temperature to 175 °C at 150 K·h^−1^; heating to 225 °C at 25 K·h^−1^; heating to 325 °C at 10 K·h^−1^; heating to 440 °C at 5 K·h^−1^, and cooling down of the furnace to room temperature by natural convection. Finally, the parts were sintered in the tubular Hobersal furnace ST 186030 (Hobersal, Barcelona, Spain) under air. The following thermal cycle was used for sintering: room temperature to 450 °C at 180 K·h^−1^, hold at 450 °C for 1 h; heating to 600 °C at 180 K·h^−1^, hold at 600 °C for 1 h; heating to 1365 °C at 300 K·h^−1^, hold at 1365 °C for 3 h, and cooling down of the furnace by natural convection. During debinding and sintering steps, specimens were placed upside-down so that the face in contact with the build platform during FFF was the air side during sintering, and no further defects were introduced.

### 2.5. Characterization of Specimens

After sintering the specimens had dimensions of approximately *T* = 3 mm, *W* = 2.85 mm and *L* = 30.75 mm. There was no significant difference between the sizes of the specimens produced with the different infill orientations. In order to determine the relative density in the green printed specimens, the density of the pelletized feedstock filaments was measured with the helium pycnometer Micrometric Accupyc 1330 (Micromeritics Instrument Corporation, Norcross, GA, USA). For each infill orientation, the five specimens produced in the first build cycle were used to determine the apparent density after printing and after sintering through the simple Archimedes’ method.

The face of the specimens in contact with the build platform, corresponding to the first extruded layer, was the side of the specimens evaluated under tensile stress in the bending tests. All the specimens were chamfered to eliminate the influence of edge defects in the perimeter [45]. The two edges of the first layer of the specimens were machined with chamfers of approximately 0.3 ± 0.075 mm.

Three-point bending (3PB) tests were performed following the standard DIN EN 843-1:2008-08 [45]. The support distance was 20 mm; the support and loading rollers had a diameter of 2.5 mm. After applying a preload of 10 N, the tests were conducted on a Zwick Z010 testing machine (Zwick GmbH & Co.KG, Ulm, Germany) at a cross-head speed of 1 mm·min^−1^ at 24 °C and approximately 56% relative humidity. The failure occurred within 6 s to 12 s. Nineteen, 22 and 20 specimens were tested for infill orientation of 0°, 90° and ±45°, respectively. The fracture stresses calculated from fracture loads and specimen dimensions were not corrected for the chamfers, leading to an underestimation of the strength values by approximately 4%. For each condition, the obtained fracture strength results were described using a Weibull distribution [46]. The parameters *σ*_0_ (characteristic strength) and *m* (Weibull modulus) of the distributions were obtained using the maximum likelihood method. No unbiasing correction was applied to the Weibull moduli obtained by this procedure.

The distribution of porosity was measured on polished sections at various locations and orientations with respect to the specimen’s long-axis. These samples were prepared from broken bent bars by polishing to a 1 μm diamond suspension finish using a semi-automated Struers RotoForce-4 and RotoPol-25 system (Struers Aps, Ballerup, Denmark). Composite micrographs consisting of six individual images each were taken at each location and orientation using an Olympus BX50 light microscope (Olympus Corporation, Tokyo, Japan) and the image alignment routine implemented in the Olympus Stream Desktop 2.2 image analysis software (Olympus Soft Imaging Solutions GmbH, Münster, Germany)

Polished sections were thermally etched in an air atmosphere for 30 min at 1400 °C to reveal the microstructure of the material. In ceramics, the strength of an individual specimen is related to the size of the most critical defect that is present. The scatter of strength is related to the size distribution of defects in the entirety of all specimens [47]. Knowledge of the types of failure causing defects that can be gained from fractography [29] can be used to understand the observed trend of strength with infill orientation. Fracture surfaces and polished sections were investigated after gold sputtering using an Olympus SZH10 stereomicroscope (Olympus Corporation, Tokyo, Japan) and a JEOL NeoScope JCM 6000Plus scanning electron microscope (Nikon Corporation, Konan, Japan).

## 3. Results

### 3.1. Properties of the Feedstock

Figure 2 shows the filament diameter over the length of the spool employed for the production of the specimens. Filament dimensions were monitored during production, and extrusion parameters were adjusted to obtain a filament with an average diameter of 1.75 mm and in the range of 1.7 to 1.8 mm. Nevertheless, uncontrolled variations of the pressure in the extruder resulted in a sudden increase or decrease of the filament diameter.

Figure 3 shows the cryofracture section of the filaments produced with the feedstock. As can be observed, the powder and binder are homogeneously distributed in a microstructure without large pores. A homogeneous and dense filament is crucial for the successful production of dense FFF components. In our experience, filaments with high porosity or an inhomogeneous microstructure have poor mechanical properties compared to dense and homogeneous filaments. Reduced mechanical properties could lead to the failure of the filament production and spooling, and even complicate the shaping by FFF [9,12,48,49]. The mechanical properties of the filaments were measured by means of tensile tests. The strength and flexibility of the filaments were quantified using the yield stress and the strain at yield, which had values of 18.8 ± 0.8 MPa and 4 ± 0.1%, respectively. The filament stiffness was quantified with the secant modulus between 0.1 and 0.3% of strain, obtaining a value of 1221 ± 84 MPa. The properties of another set of filaments produced with a binder developed in our previous study [12] and the same powder fraction and methods described in Section 3.2 were measured as a reference. In that case, the stress and strain at yield are 19.5 ± 0.5 MPa and 2.9 ± 0.2%, respectively, and the secant modulus is 1739 ± 63 MPa. Despite its lower stiffness, the higher flexibility of the filament employed in this study resulted in an easier spooling, handling and FFF processing.

The flow behavior of the feedstock is another crucial factor for a successful FFF process. The rheological measurements provide further information about the powder–binder interaction and the stability of the flow. A section of the filaments was regranulated and used in the rheological measurements with a high-pressure capillary rheometer to determine its rheological properties. Measurements were also conducted on the binder to determine the influence of the powder addition and further processing. Figure 4a shows representative curves of the pressure at the entrance of the three used dies for the feedstock and the binder.

As can be observed, the pressure increases and reaches a stable value for the binder as the piston speed is increased over time. However, the feedstock showed considerable variations at higher speeds of the piston, or what is the same, as the apparent shear rate increased. These large oscillations of pressure appeared at lower apparent shear rates as the length of the die increased. For the die with a length of 1 mm, the pressure could be measured up to an apparent shear rate of 500 s^−1^, whereas only 350 s^−1^ could be reached for the 20 mm die, and 275 s^−1^ for the 30 mm die. At those apparent shear rates, the apparent shear stress was 0.25 MPa, 0.25 MPa and 0.24 MPa for die-lengths of 10, 20 and 30 mm, respectively. During the oscillations of pressure, the material came out of the nozzle in a discontinuous periodic manner, with no material coming out of the nozzle for a short time followed by a sudden burst of material. For the binder, the maximum programmed apparent shear rate of 750 s^−1^ could be reached for all the dies employed, and the material could be extruded continuously. Due to the pressure variability, the viscosity of the feedstock could only be measured up to intermediate values, as can be observed in Figure 4b. Independently of the nozzle employed, during the measurements bubbles appeared in the extruded feedstock a few millimeters after it left the nozzle. In Figure 4c, an example of the extrudate with such bubbles is shown. Such a phenomenon could not be observed for the binder.

The thermo-oxidation behavior of the binder and the granulated feedstock filament was measured by TGA, obtaining the results shown in Figure 5. In both cases, two decomposition ramps can be observed, a ramp at low temperatures with a pronounced mass loss, and a ramp at higher temperatures with a lower mass loss rate. Each ramp corresponds to the progressive degradation of the polymers in the multicomponent binder system, as has been already observed for zirconia feedstocks for ceramic injection molding [50]. From Figure 5, it can be seen that the slope of all the ramps is more pronounced and the degradation starts at lower temperatures for the feedstock than for the binder.

### 3.2. Properties of the FFF specimens

Table 1 summarizes the theoretical and actual green mass for all the specimens produced. For the five specimens produced in the control build cycle, the mass, apparent and relative density in the green and sintered state are shown as the “control build cycle” in Table 1. The theoretical mass was calculated considering a filament with a constant diameter of 1.75 mm, the length indicated in the G-Code of each infill orientation (Appendix A) and the density of the feedstock measured in the helium pycnometer (3.296 ± 0.002 g/cm^3^). The small differences between the theoretical green mass and the actual green mass of the specimens are caused by the variability in the filament diameter (Figure 2), which results in the variability of the mass of the produced specimens. A section of the filament with a diameter higher than 1.75 mm results in overextrusion of material, whereas underextrusion occurs for filament diameters lower than 1.75 mm. Nevertheless, there is no significant difference between the actual green mass of the specimens produced with the different infill orientations.

Due to differences in filament diameter in sections employed to produce the control build cycles, there is overextrusion for the parts from the control build cycle for ±45° and 0° orientations, printed with a filament diameter higher than 1.75 mm, and underextrusion for those of the control build cycle of with 90° orientation, which were produced with a filament diameter smaller than 1.75 mm. The parts for the three control build cycles show the same mass loss of approximately 15 wt% after sintering, which is equivalent to 46.5 vol% of powder in the printed parts, as opposed to 47 vol%. This difference could be caused by the continuous underfeeding of powder during compounding by a small and systematic error in the feeding units in the compounder. The use of a slightly higher feeding rate for the powder could solve it. Still, it would require various iterative cycles that were not possible due to the limited amount of material available.

Since the simple Archimedes’ method was used to measure the density of the parts, some water could penetrate the open porosity at the surface of the specimens. Therefore, the density shown in Table 1 is the apparent density, and not the bulk density [7]. The apparent density of the feedstock was also measured, as the helium used in the pycnometer measurements could penetrate the outer pores of the pellets. Nevertheless, since the perimeter was produced with the same procedure independently of the orientation of the infill, a direct comparison of the apparent density can be established. Table 1 shows that the parts produced with 0° infill orientation are denser than those produced with ±45° orientation, and both are denser than those produced with 90° orientation. For the parts produced with 90° orientation, one cause of this difference could be the porosity produced by the underextrusion for this build cycle. Nevertheless, the trend is opposite for the parts produced with ±45° orientation, which were heavier but less dense than the ones with the 0° orientation.

As can be expected, the relative green and sintered density values follow the same trend with the infill as the apparent density. Overall, the relative green density is lower than 100%, and due to the residual porosity caused by incomplete sintering, the relative density after sintering is even lower than after printing. In order to get a better understanding of the type and size of the different pores, and the origin of them, a detailed microscopy evaluation was conducted.

The macroporosity in the sintered and fractured specimens was evaluated for cuts in different directions. In Figure 6, the results of this study are presented. The green and solvent debound specimens showed the same type of defects, as Appendix A shows. Three types of pores can be observed in Figure 6, each one in a wide variety of sizes: round pores, elongated pores and pores with an irregular shape.

The width and length of pores and their intralayer distribution are shown in the sections xy of Figure 6. These sections were obtained by polishing between 200 and 500 μm of the lower part of the specimens, i.e., the side under tensile loads in the bending tests. In all the cases, the pores shown in these sections follow the fill pattern of the extruder shown in Figure 1. In the outer part of the specimens, the pores follow the perimeter, which is the same for the three infill orientations. In the inner part, the orientation of the pores depends on the infill orientation employed. Pores are aligned in the x-direction for specimens with 0° infill orientation, and in the y-direction for specimens with 90° infill orientation. The pores inside the specimens with the ±45° infill orientation are diagonally aligned to the parts.

The distribution of pores in the different layers can be observed in the sections xz-C, xz-S and xy. These sections show the inhomogeneity in the porosity distribution in all specimens. The most pronounced difference is a reduced porosity in the outer part of the specimens than in the inner part. A low porosity can be seen on the right of the xz-C sections and the sides of the yz sections (Figure 6), and moreover, the sections xz-S have a considerably lower porosity than the xz-C. All these sections correspond to specimen sides, where 50% overlap between perimeter and infill was applied, and thus material overextrusion.

A more subtle difference is the low porosity of the lower part of the specimens compared to the rest, which can be observed in the cuts xz-C, and yz. This area corresponds to the first layers of the specimens, which are closer to the build platform. Before each build cycle, the height of the build platform was adjusted manually relative to the extrusion head. A tight adjustment was conducted to ensure proper adhesion of the specimens to the platform; this means that the actual height of the first layer was in most of the cases slightly lower than the programmed height of 0.15 mm. In this manner, there is an additional overextrusion of material in the lower part of the specimens that can prevent pores.

The xz and yz sections of Figure 6 show that the infill of the specimens has a different distribution (size and shape) of interlayer porosity depending on the plane orientation of the section with the infill orientation of specimens.

For instance, the sections xz-C are parallel to the infill orientation of the specimens with 0° orientation, which show elongated and thin pores in these sections. For the specimens with ±45° infill orientation, the plane of the section has an alternating orientation of +45° and −45° with the infill; in these sections, irregular and large pores are predominant in this case. The infill orientation of 90° is perpendicular to these sections, and small round and irregular pores are predominant.

Since section yz is perpendicular to the infill orientation of 0°, pores inside the parts with this orientation are round and small. The plane of the yz sections is diagonal to the infill orientation of ±45°, and there is an equal amount of irregular and round pores with different sizes. Since the infill orientation of 90° is parallel to the yz section, elongated pores are predominant. However, pores are considerably larger than those of the specimens with a 0° infill orientation in the xz-C sections.

The last phenomenon observed in Figure 6 is the presence of pores inside the extruded roads. These pores are distinguishable as round pores in the right specimen border in the sections xz-C, or in the two side borders of the specimens in the sections yz, which correspond to the perimeter of the specimens. The outer area of the xy sections also corresponds to the perimeter of the specimens, where more pores are distributed. In those areas, some of the pores are round, while some others are elongated and adopt an elliptical shape. For the areas corresponding to the infill of the specimens, it is difficult to determine which of the pores are inside the extruded roads and which are the pores commonly seen in the gaps between roads [8,14].

The other potential source of porosity in the specimens is the porosity which could not be removed during sintering. This kind of porosity can be regarded as an intrinsic feature of the material and is also present in similar ceramic components produced by conventional routes [51]. Attempts to remove it by variation of the sintering conditions would lead to oversintering and could cause excessive grain growth that could be detrimental for the mechanical properties [52]. Figure 7 shows the microstructure of a specimen produced with the ±45° infill orientation. Despite some micropores remaining, a homogeneous and dense microstructure can be observed, with small-sized grains (approx. 300 nm to 500 nm).

The infill orientation has a significant effect on the defects in the side of the specimens that were in contact with the build platform, which was the one under tensile loads in the bending tests. Figure 8 shows examples of the defects on this side with the different infill orientations. In the specimens with the 0° infill orientations, thin gaps can be observed between the roads. These gaps become more irregular and broader for the specimens with the ±45° infill orientation, with some large defects in between. For the specimens with the 90° infill orientation, the second type of defects is even more pronounced, and can be easily detected. Moreover, the second type of defects is only found in the inner part, which corresponds to infill roads.

Figure 9b shows the Weibull plot obtained in the three-point bending tests of the specimens produced with the different infill orientations. Schematics of the bending test configuration are shown in Figure 9a to facilitate the interpretation of the results. Table 2 summarizes the values of the characteristic strength, the average strength and the Weibull modulus. The specimens produced with the 90° infill orientation had a significantly lower characteristic strength than those specimens produced with the other infill orientations. Due to the high variability of the results, no significant difference exists between the characteristic strength of the specimens produced with the ±45° and 0° infill orientations; however, the latter exhibits higher values. When comparing the Weibull moduli of the different specimens, the specimens with the 0° orientation have the highest value, followed by those produced with the ±45° and the specimens with the 90° having the lowest average values. Nevertheless, only the difference between the Weibull modulus of the specimens with 0° orientation and that of the specimens with 90° is statistically significant.

In Table 2, the properties measured in three-point bending tests for 3 mol % yttria-stabilized tetragonal zirconia parts produced by additive manufacturing and conventional methods are shown for comparison. The strength values are higher than MEX-components produced with the same infill orientation [53]. However, the strength values are lower than for VPP and subtractively manufactured components [54]. Moreover, the strength is considerably lower than the typical value of 1200 MPa provided by the supplier [41].

Despite a large amount of porosity, the fracture surfaces exhibited fractured mirrors and thus fracture origins could be identified. All specimens failed due to the pores and defects previously described. Figure 10 shows the fracture origin of exemplary specimens with high and low bending strength. Especially for the specimens with 90° infill orientation, the hackle region which usually shows ridges and grooves radiating away from the failure origin [29] is strongly influenced by the macropores and the layerwise assembly of the specimens. Low strength specimens with 90° infill orientation failed due to defects at the road interfaces which are located on the tensile faces of the specimens (Figure 10). These typical defects are oriented perpendicular to the applied stress and are thus most dangerous. High strength specimens with 90° infill orientation did not show interfacial defects on the tensile sides but failed because of pores located within the volume, see Figure 10. The strong nonlinear trend of the data in the Weibull plot (Figure 9) may also be a hint for the existence of two different defect populations. A similar defect characteristic could be observed for the specimens with ±45° infill orientation. The interfacial defects visible on the tensile sides of the low strength specimens are oriented at an angle to the applied stress (Figure 10). They are therefore less critical than in the case of the 90° infill specimens. For the specimens with 0° infill orientation, interfacial defects which are visible on the tensile sides (see Figure 8 and section yz 0° of Figure 6) are loaded in a longitudinal direction. Failure origins for specimens with 0° infill orientation were pores at some distance from the tensile surface. For several specimens, the size of a critical defect *a*_c_ as calculated from the fracture stress *σ*_f_ using the failure criterion from LEFM [56] K_Ic_ = *σ*_f_ Y √*a*_c_ with a typical fracture toughness of K_Ic_ = 4 MPa√m for Y-TZP [52] correlates well with the observed defect sizes in Figure 10. A geometry factor, Y = 2/√π, was used for volume defects, and Y = 1.12 √π for surface defects, respectively [56].

## 4. Discussion

Despite the simplicity of the FFF equipment, the processing of ceramics by FFF is a complex process, on which different factors affect the properties and quality of the final ceramic parts. Therefore, all processing steps must be considered to determine the influence of the infill orientation on the properties of final zirconia parts.

The rheological measurements on the binder and the feedstock reveal the notable effect of the zirconia powder in the flow behavior. The expected increase of viscosity after incorporating powder (Figure 4a) is produced by the restricted mobility of the polymeric chains by the solid particles and by the interaction and friction of those particles [57]. The oscillation of the pressure during the rheology of the feedstock shown in Figure 4b has also been observed for unfilled polymers such as polyethylene, highly filled polymers and PIM feedstocks [57,58,59]. Depending on the nature of the polymeric matrix and filler fraction, different trends have been observed for this oscillation [58]. For multicomponent feedstocks such as the one employed in this study, the pressure oscillations appear only at high concentrations of powder and become more irregular as the powder concentration increases [60,61]. The fact that the oscillations start for all the nozzles at a shear stress of approximately 0.25 MPa could indicate the existence of critical shear stress after which slip of the material in the capillary wall starts [62], leading to the slip-stick effect [59]. Therefore, it can be said that high-shear rates can negatively affect the extrusion process of this material due to flow instabilities. However, it is not expected that the FFF of ceramics reaches those high-shear rates due to the high viscosity of the material, which could lead to problems like buckling or shearing of the filament.

It is also not clear whether the appearance of bubbles of the feedstock coming out of the rheometer (Figure 4c) is related to the pressure oscillations, especially since the bubbles appeared in the whole range of shear rates evaluated, whereas the oscillations started at higher rates. The origin of the bubbles could be related to moisture evaporation at high temperatures [26] or the degradation of binder ingredients at high temperatures catalyzed by moisture, powder, or the combination of both. The polar surface of the zirconia powder and the hydroxyls groups on it [63] could promote the absorption of water [64]. Moreover, water in the feedstock could explain the lower degradation temperatures for the feedstock than for the binder in the TGA (Figure 5) [65]. However, it is surprising that no bubbles appeared in the feedstocks filaments during their production at temperatures up to 245 °C (as shown in Figure 3). Since pelletized feedstock filaments were used in the rheology measurements and the TGA, the additional heating and shearing cycles on the material could contribute to the degradation of the material. At that step of the process, polymers have been melt-mixed to obtain the binder, compounded with the powder to get feedstock, extruded to make filament, and finally extruded during the characterization or the shaping by FFF.

Moreover, the pores inside the extruded roads could be another manifestation of the phenomena described in the previous paragraph. The origin of these pores has been attributed to the existence of previous porosity in the filament [21]. In our case, since the filaments employed in this study have a homogeneous and dense microstructure with no pores (Figure 3), it can be stated that the pores inside the extruded roads appear during the shaping of the parts by FFF and not in the previous steps of compounding and filament production. Pores inside perimeter roads can be seen as round cavities, which can also be observed in the bulk of specimens, but also as elliptical pores in the perimeters in the xy sections (Figure 6). For the inner region of the specimens, it is not possible to differentiate the pores inside the roads with the gaps between the roads.

The gaps between roads are a direct consequence of the building strategy not only of FFF [8,14,31] but also of other MEX processes such as piston-based [33,34] or screw-based [31,35] systems. These gaps can appear as interroad defects, which are elongated voids in between the extruded roads and parallel to them, or as subperimeter voids, which are produced by the change of direction of the infill when it reaches the perimeter [8,14]. In this study, there is an overlap of 50% between perimeters and infills, which effectively reduces or eliminates the subperimeter voids [8].

In previous studies dealing with the production of metallic and ceramic components, the interroad defects have been detected as long voids running parallel to the roads across the whole section of the parts [14,17]. Such defects cannot be observed in any of the sections evaluated in Figure 6. However, there are small elongated voids observed in the cuts parallel to the road deposition (xz-C for 0° and yz for 90°) and with small size in the cuts of Figure 6 perpendicular to the road deposition (yz for 0° and xz-C for 90°). Nevertheless, some of these voids could be the elliptical pores inside the roads discussed previously. Since the specimens produced with the ±45° infill orientation have roads perpendicular to those of the previous layer, the shape of the interroad defects is different than the shape of these defects for the other orientations, and no direct comparison can be made.

The inconsistency in the filament diameter in this study (Figure 2) could promote the irregularity of the interroad pores in two ways. The direct effect is that a filament section with small or large diameter results in an uncontrolled under- or overextrusion of material [8,26,27]. The second effect is the flow fluctuations by the filament diameter variability, which changes not only the amount of material extruded but also the grip of the rollers on the filament and the force applied on the molten feedstock in the nozzle [8,27,66]. Appendix A shows the variation of the mass of the specimens in the different build cycles as a result of the filament diameter; however, no direct correlation between the mass of the specimens and their bending strength could be observed (Appendix A).

Two sources of overextrusion can be found in this study: the programmed overextrusion caused by the perimeter-infill overlap and the uncontrolled overextrusion by the changes in the filament diameter. In principle, the overextrusion of material is an effective way to reduce the porosity in FFF parts [67,68]. Nevertheless, an excessive overextrusion leads to a “flooding” of material between roads and structures such as the one shown in Figure 11a [8,68]. Figure 11b shows the origin of the overextrusion defects, which can eventually result in the apparition of new types of defects, as shown by Costa et al. [68]. The shape and volume of these defects are different depending on the orientation between the roads of consecutive layers, as shown in Figure 11c. With the 0° and 90° infill orientations the roads of the new layer are parallel to those already deposited, resulting in the defects reported by Costa et al. In the specimens with the ±45° infill orientation, the new roads are perpendicular, which leads to new types of defects.

The defects in the lower side of specimens (Figure 8) are directly correlated with the length of the roads for each infill orientation. In Figure 1, it can be observed that long roads are used in the 0° infill orientation, whereas the roads are shorter for the ±45° infill orientation and especially for the 90° infill orientation. Longer roads result in more time for cooling of the material, which in the bulk of the specimens could be detrimental since the bonding between the roads is reduced [8]. However, the situation is different in the first layer of the specimens. If the deposited roads in contact with the build platform are not cold enough, the material could be scrapped away in some areas, resulting in the irregular and large defects observed in the specimens with the ±45° and 90° infill orientations in Figure 8.

In three-point bending tests, all specimens failed due to the pores and defects previously described, which were generated during the FFF of the parts. The FFF-induced defects (Figure 6 and Figure 8) are more prominent and bigger than the intrinsic material defects (Figure 7), which explains the generally low strength compared to traditional and VPP manufactured Y-TZP [41,54,69,70]. In the specimens with the 90° und ±45° infill orientations, different defect types are active. The large defects in the tensile surface (Figure 8) cause the fracture of the low strength specimens, and the origin of the fracture in the high strength specimens are the pores in the core observed in Figure 6. This porosity results in low Weibull moduli and high strength scatter for the specimens with ±45° infill orientation and especially those with 90°. In the specimens with 0° infill orientation, the thin interroad defects on the tensile surface did not cause failure due to the low number of roads used to build the first layer and the favorable orientation of their interfaces concerning the applied stress. In these specimens, only one defect population was responsible for failure: The volume defects (pores) that are equal in size as in other orientations and appear at different locations in the cross-section, in some cases at a rather long distance from the tensile faces of the specimens (Figure 10). As a consequence, it results in higher characteristic strength and a lower scatter of strength for the 0° infill orientation specimens.

## 5. Conclusions

In this study, the influence of the infill orientation on the properties of zirconia parts produced by FFF was evaluated. Bending bars with infill orientations of 0°, ±45° and 90° were produced, and the properties of ceramic feedstock and sintered parts were measured.

The incorporation of the powder into the organic binder results in significant changes in the rheological properties and thermal degradation behavior. Pressure oscillations appeared during the rheological measurements of the feedstocks at a shear stress of approximately 0.25 MPa, whereas no oscillations were recorded for the binder. Moreover, gases formed during the extrusion process result in pores inside the roads during the shaping by FFF.

FFF shaping causes three defect types: interroad defects, defects due to material under- and overextrusion, and defects caused by the shearing-off by the nozzle of the already deposited material in the first layer. All of these defects are promoted by the variability of the filament diameter, and their shape and orientation are affected by the infill orientation, resulting in different bending strength of the evaluated specimens. The shape of the defects due to excessive overextrusion is different if the roads of the next layer are parallel or perpendicular to the roads of the previous layer. Finally, the shearing-off of the deposited material in the first layer results in defects for the ±45° and 90° orientations.

The strength of the parts was evaluated in the as-sintered state. The bending behavior of the parts is primarily influenced by the quality of the first layer. Interfaces that are oriented normal to the applied stress (as in 90° and to some extent in ±45° infill orientation), bad leveling of the bed, variations of the filament diameter and the shearing-off the deposited material result in high variability of the strength values and low Weibull modulus. The bending strength of parts printed with 0° orientation shows a smaller dependence on these defects than the strength of those parts with 90°. On the other hand, pores inside the specimens are causing failure if the tensile side is free of defects.

Therefore, the loads applied to FFF ceramic parts during their service have to be considered during their production. When possible, the infill roads must be oriented in parallel to the tensile loads. In order to produce dense and strong components by FFF, the defects inside and between the roads must be avoided. The use of filaments with tight dimensional tolerances and the use of controlled overextrusion are known to reduce the interroad defects. In future investigations, the influence of the powder content and the moisture on the degradation of the binder components must be studied to minimize intraroad porosity.

## Figures and Tables

**Figure 1 materials-13-03158-f001:**
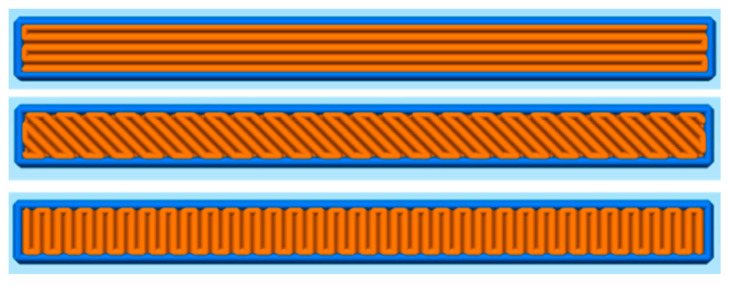
Fill pattern of perimeters (blue) and infills (orange) of one layer for specimens of each of the infill orientations, from top to bottom: 0°, ±45° and 90°.

**Figure 2 materials-13-03158-f002:**
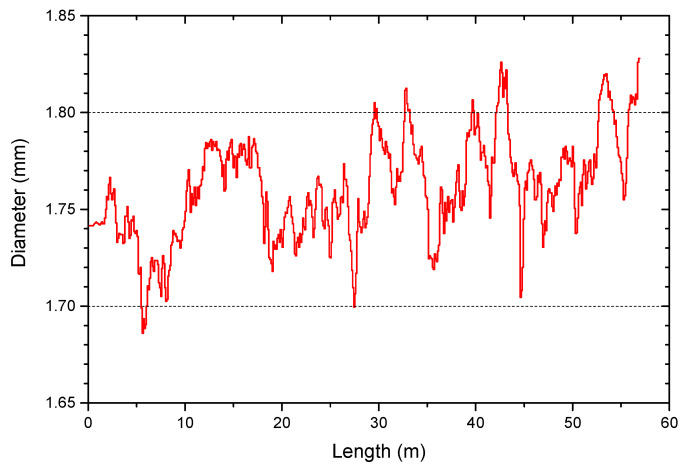
The diameter of the spool employed in the production of the fused filament fabrication (FFF) specimens.

**Figure 3 materials-13-03158-f003:**
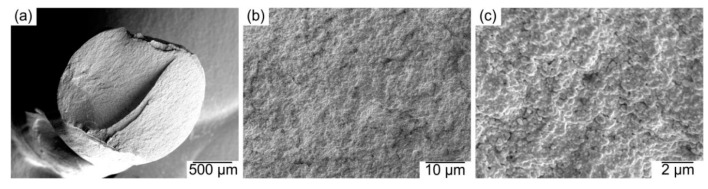
Morphology of the cryofractured feedstock filament at: (**a**) 100 times magnification; (**b**) 5000 times magnification and (**c**) 20,000 times magnification.

**Figure 4 materials-13-03158-f004:**
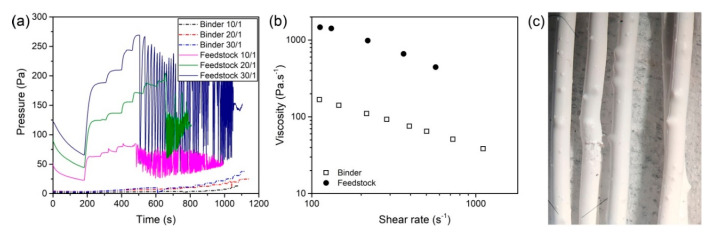
(**a**) Representative pressure curves measured for the binder and feedstock with the three nozzle geometries; (**b**) shear viscosity as a function of the shear rate for the binder and feedstock after applying the Bagley and Weissenberg–Rabinowitsch corrections; (**c**) feedstock rods collected during the rheological measurements.

**Figure 5 materials-13-03158-f005:**
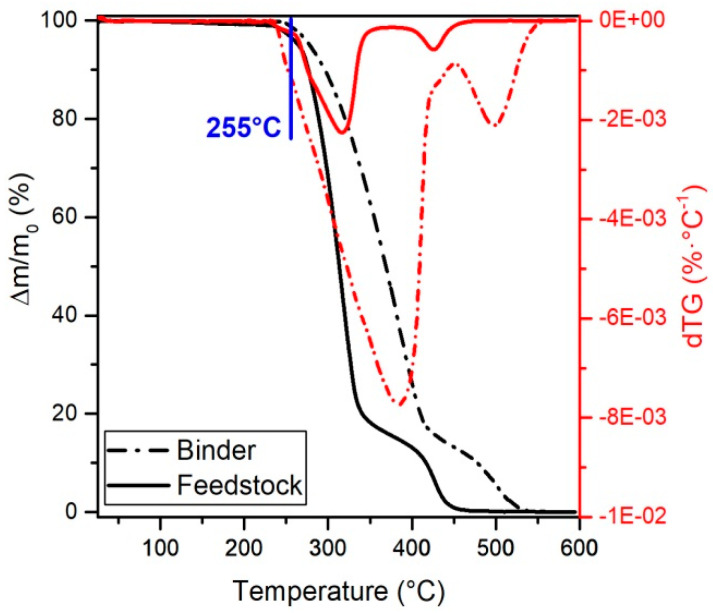
Representative curves of the amount of binder left in samples as temperature increases, and of the derivative of the binder left in the samples at different temperatures for the binder and feedstock. The nozzle temperature (255 °C) used in the FFF shaping is indicated in blue.

**Figure 6 materials-13-03158-f006:**
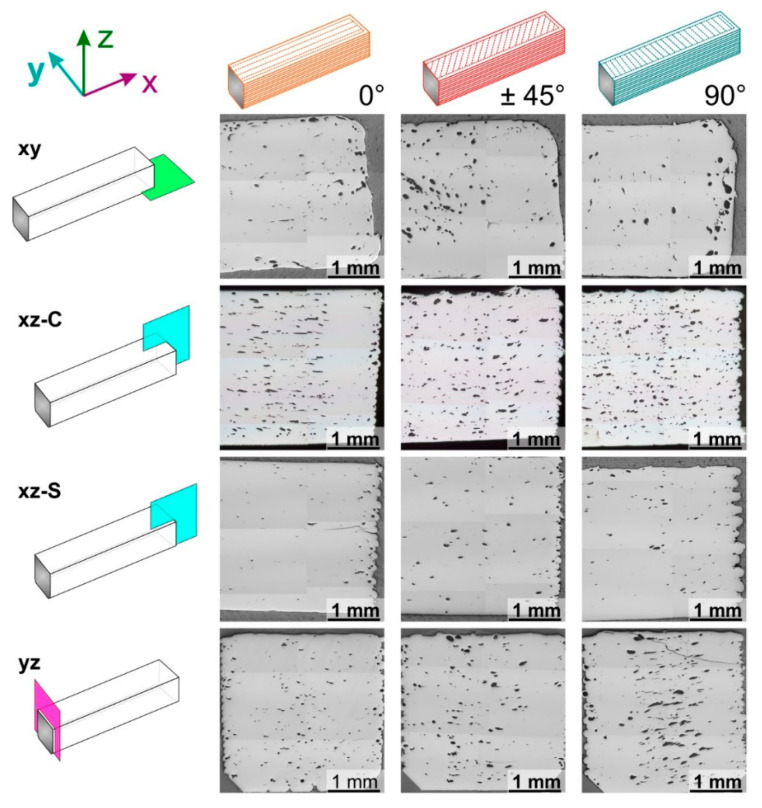
Sections of sintered fractured bars with the three infill patterns. The location and orientation of the imaged plane are indicated in the leftmost column with the directions parallel to the plane used to designate it. The zones studied are the center (C) or the side (S). The fracture surface is marked with a dark gray color in the schematic representation.

**Figure 7 materials-13-03158-f007:**
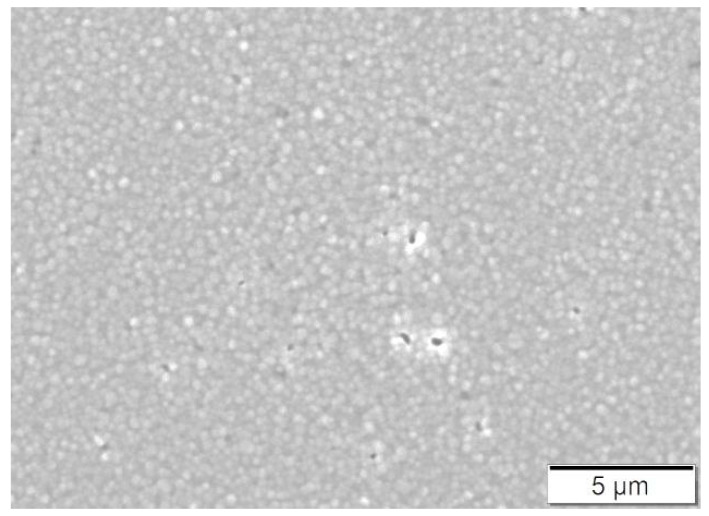
Typical microstructure of a sintered specimen.

**Figure 8 materials-13-03158-f008:**
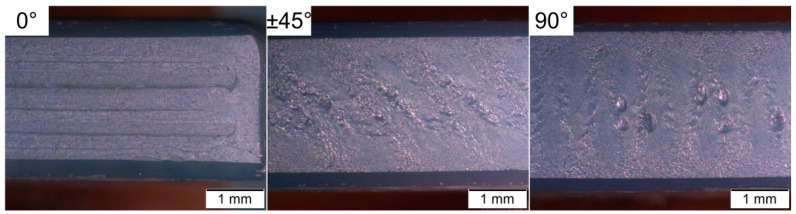
Underside of the sintered specimens produced with the different infill orientations after chamfering.

**Figure 9 materials-13-03158-f009:**
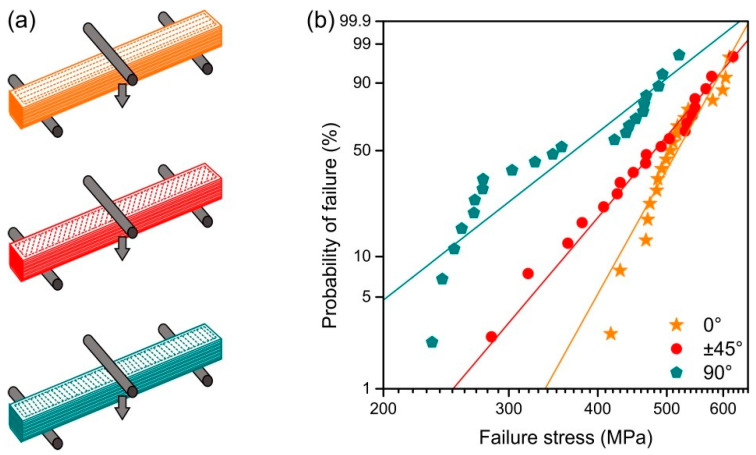
(**a**) Schematics of the bending tests for specimens produced with the three infill orientations; (**b**) Weibull plot of the specimens produced with the different infill orientations.

**Figure 10 materials-13-03158-f010:**
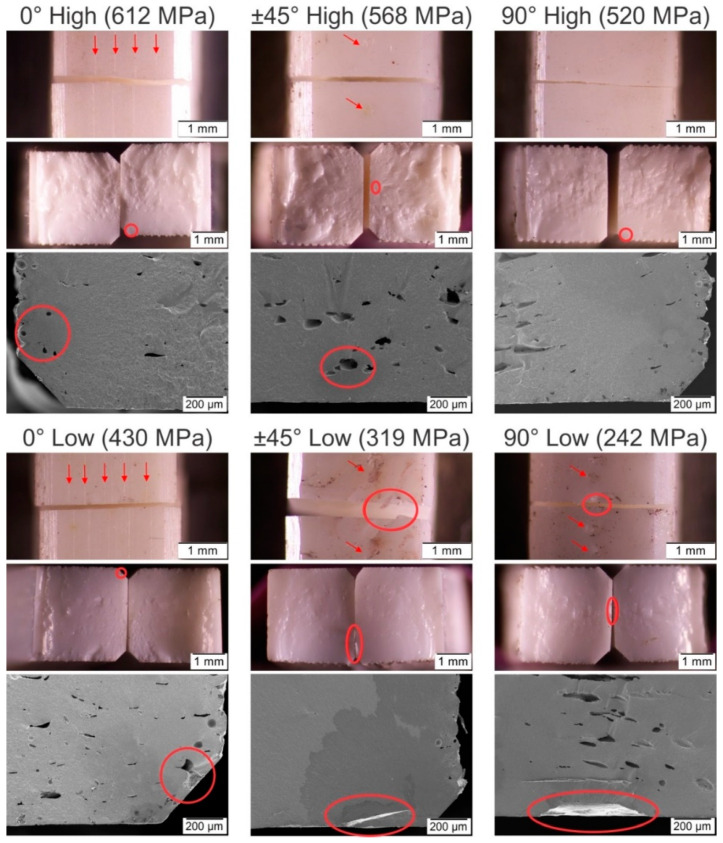
Tensile surfaces and fracture surfaces of specimens with high and low strength and Influence of the defects on the first layer produced during shaping by FFF for the specimens with high and low strength produced with infill orientations of 0°, ±45° and 90°. Arrows mark periodic defects between the roads produced during FFF. Locations of the fracture origins are marked with circles and the defects in the tensile surface are marked with arrows.

**Figure 11 materials-13-03158-f011:**
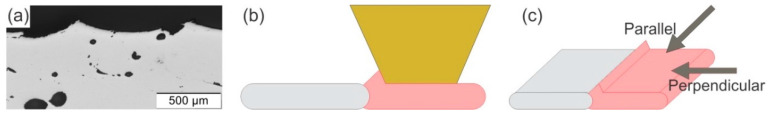
(**a**) Effect of excessive overextrusion on the top extruded layers; (**b**) schematics of the origin of the overextrusion defects by the extrusion of a new road of hot material (in red) over a previously deposited road of cold material (in gray); (**c**) schematics of the material deposition on top of a road with overextrusion if the roads of the new layer are parallel or perpendicular to it.

**Table 1 materials-13-03158-t001:** Theoretical and actual green mass (m_g_) for all the specimens produced with the different orientations and for the five parts of a control build cycle for each orientation: green (m_g_) and sintered mass (m_s_), mass loss (Δm), green (ρ_g_) and sintered (ρ_s_) apparent density, and relative green (ρ_rg_) and sintered (ρ_rs_) density. The average and standard deviation values are provided for all the parameters.

	Theoretical	Actual	Control Build Cycle						
Infill Orientation	m_g_ (g)	m_g_ (g)	m_g_ (g)	m_s_ (g)	Δm (wt %)	^1^ ρ_g_ (g/cm^3^)	^1^ ρ_s_ (g/cm^3^)	ρ_rg_ (%)	ρ_rs_ (%)
0°	1.702	1.706 ± 0.029	1.723 ± 0.01	1.464 ± 0.01	15.06 ± 0.07	3.254 ± 0.003	5.943 ± 0.013	98.73 ± 0.08	98.23 ± 0.22
±45°	1.703	1.72 ± 0.041	1.77 ± 0.007	1.487 ± 0.042	15.08 ± 0.03	3.202 ± 0.024	5.878 ± 0.034	97.15 ± 0.71	96.97 ± 0.64
90°	1.694	1.72 ± 0.034	1.642 ± 0.004	1.396± 0.003	15 ± 0.06	3.176 ± 0.007	5.797 ± 0.015	96.35 ± 0.21	95.82 ± 0.26

^1^ The green apparent density is calculated using the apparent density measured for the feedstock in the helium pycnometer (3.296 ± 0.002 g/cm^3^), and the sintered density is calculated using the density provided by the powder supplier as reference (6.05 g/cm^3^).

**Table 2 materials-13-03158-t002:** Weibull modulus (m), characteristic bending strength (σ_0_) and average bending strength (σ) of the specimens produced with the three infill orientations. For comparison, 3 mol % yttria-stabilized tetragonal zirconia parts produced by additive manufacturing material extrusion (MEX), vat photopolymerization (VPP) and subtractive methods are shown. A 90% confidence intervals are given in square brackets.

Technology	Infill Orientation	m (-) [90% CI]	σ_0_ (MPa) [90% CI]	σ (MPa)
	0°	9.9 [6.7–12.6]	537 [514–562]	512 ± 55
FFF (MEX)	±45°	6.5 [4.5–8.3]	508 [477–543]	473 ± 90
	90°	4.3 [3.1–5.5]	404 [368–443]	366 ± 99
3D Gel Printing (MEX) [53]	0°	28 ^1^	462 ^1^	450 ± 20
Digital Light Processsing (VPP) and polishing [54]	-	9.3 [6.7–12.3]	1066 [1031–1101]	1013 ± 126
Subtractive method [54]	-	12 [8.7–15.8]	1206 [1173–1239]	1158 ± 114

^1^ Values calculated from the average and standard deviation of the bending strength [55].

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
