# Peer review of "Influence of the Infill Orientation on the Properties of Zirconia Parts Produced by Fused Filament Fabrication"

_materials, 2020, doi:10.3390/ma13143158_

Round 1

Reviewer 1 Report

It is a nice contribution, some improvements are possible specially comparing the produced samples with traditional manufacturing process  and describing in more details how anisotropic ( analysing performance in the different directions)  are the fabricated samples.  

Some comments

 -In the experimental section is not clear what “is confidential”.  The nozzle temperature should be specified in the figures.

- Filament diameter alter sintering should be marked in figure 6.  Relative density of the produced materials presented in figure 6 should be given.

- In figure 7 the grain are not clearly visible. A higher resolution  image should be provided.

- In figure 9 the a schematic of the  sample pattering orientation respect to the testing direction should be added.

- Reference values for isotropic bars produced by traditional pressing with and without binder   should be added in table 2 and discussed in the text.

Typos “pores insidethe specimens”.

Author Response

It is a nice contribution, some improvements are possible specially comparing the produced samples with traditional manufacturing process  and describing in more details how anisotropic ( analysing performance in the different directions)  are the fabricated samples. 

Some comments

The authors thank the reviewer for their positive evaluation of our work. We appreciate the time and effort that the reviewer put in reviewing our manuscript. We believe that the comments have resulted in a significant improvement of the manuscript. Below a detailed response to each comment is given. All amendments and corrections to the original submission are highlighted with the track changes function of word in the uploaded manuscript.

Regarding the description of the anisotropy in the samples, no more information can be added since it is out of the scope of the paper. The investigation of the anisotropy of FFF ceramic parts can be found in the works of Iyer et al. for silicon nitride (reference 17 of this manuscript) and Conzelmann et al. for alumina (reference 19 of this manuscript). Therefore, it was not decided to include it in this investigation

Point 1: In the experimental section is not clear what “is confidential”.  The nozzle temperature should be specified in the figures.

 Response 1: Thank you for this comment. The polymer binder components and their fractions cannot be disclosed due to confidentiality reasons. The nozzle temperature has been added to the TGA graphs in Figure 5

Point 2: Filament diameter alter sintering should be marked in figure 6.  Relative density of the produced materials presented in figure 6 should be given.

Response 2: We agree that the information requested by the reviewer would be of interest in the discussion of Figure 6. However, it is not possible to provide it because of the following reasons:

Due to the variability of the filament diameter, there was an uncontrolled extrusion of material. Because of the uncontrolled extrusion, the thickness and width of the extruded roads cannot be accurately distinguished to be marked correctly in Figure 6. Therefore, we considered to be misleading to add a fictive extrusion road in Figure 6.

Due to the long time required for measuring the density of the specimens on a precise manner, only the density of a control batch was measured for each infill orientation as indicated in the experimental section. Therefore, the measured density cannot be attributed to the specimens fractured during testing and shown in Figure 6.  

As previously mentioned, The changes in the filament diameter result in an uncontrolled extrusion of material which results in a large variability in the porosity and density of the specimens. Therefore each specimen is unique and has a unique porosity. Because of this variability, it is not possible to generalize the influence of the infill orientation in the density and porosity of the specimens evaluated in this publication. For these reasons porosity and density analyses were removed from preliminary versions of this manuscript. In future studies, we will try to quantify in an appropriate manner these parameters.

Point 3: In figure 7 the grain are not clearly visible. A higher resolution image should be provided.

Response 3: The grain size in the sintered is below 1 μm and unfortunately due to technical limitations of the available SEM equipment no higher-magnification and resolution image can be provided in a timely manner. To facilitate the visualization of the microstructure, the contrast and brightness of Figure 7 has been modified.

Point 4: In figure 9 the a schematic of the sample pattering orientation respect to the testing direction should be added.

Response 4: Thank you for this comment. Sketches to clarify the testing direction with respect to the infill orientation were added to Fig. 9.

Point 5: Reference values for isotropic bars produced by traditional pressing with and without binder should be added in table 2 and discussed in the text.

Response 5: Thank you very much for this comment. Following this recommendation and the comment of reviewer 3, Table 2 has been updated with properties measured in additively and conventionally manufactured components. The typical strength value of 1200 MPa provided by the supplier has been also included in the text.

Point 6: Typos “pores inside the specimens”.

 Response 6: Thank you for this comment. The term “pores in the specimens/parts” has been substituted by “pores inside the specimens/parts”

Reviewer 2 Report

Manuscript presents experimental results of impact of the infill orientation on the properties of zirconia parts produced by Fused Filament Fabrication. Procedure of a sample preparation, fabrication and the bending test are properly described. Contains plenty of technical details allowing for replication of the procedure. Manuscript contain some new and important for industry information concerning impact of infill orientation on mechanical strength. Manuscript is well written. Manuscript needs only minor corrections to be recommended for publication in Materials.

Particular remarks:

  1. Line 212. Please indicate deflection at failure instead of the time of displacement.
  2. Lines: 208-2018. Please indicate the number of performed bending tests.
  3. Examples of plot of the stress-deflection for different orientations of the infill would be very helpful in survey of results.

Author Response

Manuscript presents experimental results of impact of the infill orientation on the properties of zirconia parts produced by Fused Filament Fabrication. Procedure of a sample preparation, fabrication and the bending test are properly described. Contains plenty of technical details allowing for replication of the procedure. Manuscript contain some new and important for industry information concerning impact of infill orientation on mechanical strength. Manuscript is well written. Manuscript needs only minor corrections to be recommended for publication in Materials.

The authors thank the reviewer for the positive evaluation of our work. We appreciate the time and effort that the reviewer put in reviewing our manuscript. Below a detailed response to each comment is given. All amendments and corrections to the original submission are highlighted with the track changes function of word in the uploaded manuscript.

Particular remarks:

  1. Line 212. Please indicate deflection at failure instead of the time of displacement.

Response 1: During ceramic strength tests, the specimens usually deform only elastically. In strength testing according to EN 843-1 measurement of deflection is not required and thus has not been done. Since the strength of ceramics depends on the loading rate, the time to failure is prescribed to be between 5 – 15 s and it should be reported.

  1. Lines: 208-2018. Please indicate the number of performed bending tests.

Response 2: The numbers can be found in the text in line 213: “19, 22 and 20 specimens were tested for infill orientation of 0°, 90° and ±45°, respectively”

  1. Examples of plot of the stress-deflection for different orientations of the infill would be very helpful in survey of results.

Response 3: During ceramic strength tests, the specimens usually deform only elastically. In strength testing according to EN 843-1 measurement of deflection is not required and thus has not been done. On one hand, measurement of the deflection directly on the specimen would possibly influence test results (and damage fracture surfaces preventing proper fractographic analysis), the crosshead displacement on the other hand does not give information because the ceramic specimens are usually much stiffer than the load cell and the test fixtures and deform a lot less that other parts in the load train.

Reviewer 3 Report

The manuscript is well organized and is of great scientific soundness.

There is one suggestion for improving the quality of the article:
-to compare the characterictics of the fabracated specimens to others known in scientific
literature

Author Response

The manuscript is well organized and is of great scientific soundness.

 The authors greatly appreciate the positive evaluation of our work. We believe that the comment has resulted in a significant improvement of the manuscript. All amendments and corrections to the original submission are highlighted with the track changes function of word in the uploaded manuscript.

There is one suggestion for improving the quality of the article:

-to compare the characterictics of the fabracated specimens to others known in scientific literature:

Response: In Table 2 the results obtained for 3mol% yttria stabilized zirconia produced by other technologies and tested with three point bending tests have been incorporated for comparison. The typical bending strength value provided by the supplier has been also included in the text (line 444 of the corrected manuscript).
